# A Versioned Unified Graph Index for Dynamic Timestamp-Aware Nearest Neighbor Search

## Abstract

We present TiGER (Time-Integrated Graph for Efficient Retrieval), a novel approach for performing fast time-aware approximate nearest neighbor searches on dynamic vector datasets with flexibility over any possible time range. Our proposed algorithm builds and maintains a unified graph for all vectors by leveraging an index structure based on integrated versioned connectivity, allowing arbitrary time intervals to be queried directly on the unified graph without having to traverse invalid vectors. This forgoes the need for post-search filtering or merging, or separate graphs for each possible composite range. Empirical evaluations show that our method attains up to a 5x improvement in queries per second (QPS) without compromising accuracy over baselines based on filtering or per-time-segment sub-graphs. We believe that this method will enable efficient temporal analysis across evolving datasets in real-time recommendation systems, log analysis, and any scenario requiring fast similarity search over dynamic, time-segmented data.

## 1 Introduction

As the volume of textual data continues to expand at an unprecedented rate, efficient and accurate text retrieval has become a cornerstone for numerous online applications, including Retrieval-Augmented Generation (RAG) (Fan et al., 2024; Gao et al., 2023) and news fact-checking (Capuano et al., 2023; Liao et al., 2023). The ability to quickly retrieve relevant information is particularly critical in the context of large-scale corpora, where similarity search serves as a foundational mechanism for modern information retrieval systems. Its significance has grown with the rise of Large Language Model (LLM) workflows, where effective retrieval underpins tasks ranging from contextual augmentation to real-time query generation (Chen et al., 2022; Shorten et al., 2021; Wu et al., 2019).

To meet the demands of fast and accurate query processing, Approximate Nearest Neighbor (ANN) search has emerged as a promising strategy (Macdonald & Tonellotto, 2021; Tu et al., 2020; Xiong et al., 2021). By tolerating a small margin of error, ANN techniques achieve significant speedups, enabling rapid nearest-neighbor queries even in massive datasets. Among the various ANN methods developed (Cai, 2021; He et al., 2019; Jégou et al., 2011; Ram & Sinha, 2019), graph-based techniques have consistently demonstrated superior performance, excelling in key metrics such as recall and query time across a range of benchmarks (Fu et al., 2019; Li et al., 2020; Morozov & Babenko, 2018; Wang et al., 2021). This advantage stems from their ability to capture local neighborhood relationships, making graph-based ANN an indispensable tool for large-scale retrieval.

**Similarity Search with Time Constraints.** However, in many real-world applications, straightforward similarity search is insufficient. We often need to retrieve *topk* results under specific constraints—known as Range-Filtering Approximate Nearest Neighbor Search (RFANNS) (Zuo et al., 2024)—such as categories, keywords, or temporal restrictions (Engels et al., 2024; Kovacs et al., 2024; Zhang et al., 2022). Time-based constraints are particularly common, such as fetching news articles or social network posts within a given timeframe (Awao et al., 2023; Wang et al., 2022a) or w.r.t. periodic trends. (Bertrand et al., 2013; Golder & Macy, 2011)

Most graph-based ANN methods are not designed for such filtering and typically use post-filtering or pre-filtering (Dilocker, 2021; Xu et al., 2024): (1) *Post-filtering* retrieves topk without constraints and filters results afterward, leading to inefficiencies under tight constraints due to excess candidates. (2) *Pre-filtering* tailors the initial index to the constraint, but constructing or maintaining graphs for all possible filters is impractical. Dynamic graph construction for each query would be computationally

costly, and maintaining per-timestamp graphs and performing searches individually across the relevant graphs minimizes search cost, but requires expensive merging and ordering operations at query time.

**Challenges.** Existing approaches (Cai et al., 2024; Gollapudi et al., 2023; Xu et al., 2024; Zuo et al., 2024) address constrained ANN search but often face challenges with dynamic updates, requiring extensive graph rebuilding, or are designed for contiguous ranges, making them less suitable for more complex or fragmented constraints. In dynamic or evolving datasets, where queries and constraints frequently shift, these limitations necessitate a solution that can flexibly and efficiently integrate temporal filtering without heavy reconstruction or excessive search overhead.

**Our Approach.** We propose a unified index that seamlessly integrates time constraints into similarity search without requiring multiple graphs or extensive post-processing. Specifically, for any query vector $q$ and timestamp set $T_q = \{t_1, t_2, ...t_n\}$, whether contiguous (e.g., $[t_a, t_b]$) or disjoint (e.g., $\{t_1, t_3, t_9\}$), our method retrieves the top-$k$ approximate neighbors matching any timestamp in $T_q$ directly during search.

Our core structure is a versioned proximity graph where each node represents an embedding with temporal metadata. Nodes track active time periods, and edges are annotated with validity ranges, allowing efficient time-constrained traversal without needing graph modifications for each query. Additionally, to guarantee full reachability without costly reconstructions, we maintain dynamic predecessor links that adapt as new nodes are inserted. This structure enables direct traversal without unnecessary connections or broken graphs for any specified time range (continuous or not), eliminating the need for post-filtering or maintaining multiple subgraphs while remaining scalable for dynamic datasets.

**Contributions.** We summarize our main contributions as follows:

- We propose TiGER (Time-Integrated Graph for Efficient Retrieval), a unified graph-based ANN framework that supports arbitrary temporal filtering during search while enabling dynamic updates.
- We introduce a dynamic edge management mechanism that preserves graph connectivity across time, minimizing reconstruction costs.
- We design an integrated sparse edge database to efficiently aggregate edge information for broad or continuous time filters, improving search speed.
- We experimentally show that TiGER achieves up to a 5x improvement in queries per second (QPS) while maintaining comparable or superior recall compared to pre- and post-filtering baselines.

## 2 PRELIMINARY

### 2.1 SIMILARITY SEARCH WITH PROXIMITY GRAPHS

Due to their effectivenes in key metrics such as recall and query speed on many datasets (Fu et al., 2019; Li et al., 2020; Morozov & Babenko, 2018; Wang et al., 2021), Proximity graphs have emerged as a cornerstone option for efficient ANN search. These graphs leverage the spatial relationships between data points, constructing a graph where edges connect nearby vectors. During queries, traversal of this graph allows retrieval of neighbors with a fraction of the computational cost of exhaustive searches, making proximity graphs ideal for large-scale datasets. However, traditional proximity graphs are inherently designed for unconstrained searches, which limits their ability to handle filtered ANN tasks. Incorporating constraints, such as time ranges or categories, typically requires preprocessing (to construct filtered subgraphs) or postprocessing (to filter results after retrieval), which introduce inefficiencies (Xu et al., 2024). When filters are applied after retrieval, unnecessary candidates are traversed which wastes computations and may lead to suboptimal performance when the selectivity of the filter is high, i.e., very small number of the retrieved candidates satisfy the constraints. Conversely, pre-filtering often necessitates either on-the-fly graph construction for a given filter (as maintaining all separate graphs for all possible filters at all times is clearly impractical) or a recombination process after searches on multiple timestamp ranges.

### 2.2 PERSISTENT DATA STRUCTURES

Persistent data structures retain multiple versions of data, allowing access to historical states without duplication (Driscoll, 1989). In the context of time-based constraints, persistent data structures

provide an elegant solution for managing data across temporal dimensions (Lenhof & Smid, 1994). By encoding the temporal validity of nodes and edges, a persistent proximity graph can conserve historical data while allowing for continuous updating. Our approach applies this concept of persistence to maintain a unified, versioned graph structure. Each node and edge is annotated with metadata tracking their active states across time, ensuring the graph supports queries for any arbitrary time range. Instead of creating separate graphs for each time slice, the persistent graph enables direct traversal using the relevant versions of nodes and edges. This also allows for seamless handling of dynamic workloads, where new data points and temporal constraints are continually introduced.

## 3 TIGER FRAMEWORK

In this section, we present the TiGER framework, which employs a unified graph index to seamlessly integrate temporal constraints into its structure. This is achieved through three interconnected components: the graph index construction process (Section 3.1) establishes a single, versioned graph where nodes and edges are annotated with temporal metadata, ensuring connectivity across arbitrary time ranges. A versioning mechanism (Section 3.2) tracks the evolution of nodes and edges over time, enabling efficient retrieval without redundant reconstructions. Finally, an edge database (Section 3.3) enhances query performance for contiguous ranges by precomputing and aggregating edge information over continuous timeframes.

### 3.1 GRAPH INDEX CONSTRUCTION

TiGER builds a unified, versioned graph structure where each node represents an embedding with associated temporal validity. Unlike static indexing schemes that require separate structures for each time slice or complex post-processing, TiGER maintains a single, incrementally updated graph. This allows time-based ANN queries to operate directly on the unified index, eliminating the need for filtration or merging.

This process is illustrated in Figure 1, which shows the insertion of five vectors at a single timestamp into an initially empty graph. We represent the evolving dataset as a directed graph $G = (V, E)$, where each vertex $v \in V$ corresponds to a vector embedding $\mathbf{x}_v \in \mathbb{R}^d$. Each vertex is assigned a timestamp $t_v$ at which it was inserted, and maintains a set $active(v)$ that records all timestamps during which the vertex is considered active (i.e., eligible for inclusion in queries constrained to that time). We also define a variable $l_e \in \mathbb{N}$, which dictates the maximum number of outgoing edges that an edge can have for a single timestamp.

Each edge $e_{uv} = (u, v) \in E$ has an associated timestamp $t_{e_{uv}}$ indicating when the edge was created. Additionally, each vertex stores a record of its outgoing edge set whenever it changes. If the outgoing edges of a vertex $v$ change at timestamp $t_n$, we record this edge state as $v_{t_n}$. For any timestamp $t_i$ satisfying $t_n \leq t_i < t_m$, where $t_m$ is the next timestamp where the edges for $v$ change, the outgoing edges of $v$ at time $t_i$ are defined as $edge(v_{t_i}) = v_{t_n}$.

Two variables govern the maintenance of temporal connectivity and edge balance during insertion:

- $prev(v)$ stores a parent node that is guaranteed to have an outgoing edge to $v$. This allows a path to be reconstructed from the origin to any node $v$ by following a chain of backward pointers.
- $push(v)$ tracks the number of outgoing edges added from $v$ specifically for the purpose of connecting it to newly inserted nodes (outside of the initial greedy search connections). $push(v) \leq l_e$ is enforced to guarantee that no edge added by said process is pushed out.

The insertion process is detailed in Algorithm 1. When a new vertex $v$ is inserted, a greedy search is performed (regardless of timestamp) to find the $l_e$ closest nodes, and outgoing edges are added from $v$ to each of them. This process is in essence the same as that seen in standard proximity graphs (Zhao et al., 2020).

Next, we identify a suitable preexisting node to assign as $prev(v)$ and ensure it has an edge to $v$. Among the candidates from the greedy search, we select a node $v_k$ that has not exhausted its edge budget (i.e., $push(v_k) < l_e$). If no such node is available, a secondary greedy search is performed to find a nearby node that can accommodate an additional edge. If necessary, we remove the oldest existing edge not inserted by a previous reconnection from that node, insert the edge $(v_k, v)$, assign $prev(v) = v_k$, and increment $push(v_k)$.

**Algorithm 1** Insertion into the versioned graph

**Input**: Graph index $G$; current timestamp $t_c$; vector to insert $v_i$; origin $v_o$; edge limit for timestamped node $l_e$
**Output**: Updated index $G'$

1. Perform a timestamp-blind greedy search on $G$ beginning at $v_o$ to obtain $topk = \{v_1, v_2, ...\}$, a list of $l_e$ closest nodes to $v_i$ (ascending distance).
2. Add edges $(v_i, v_k) \, \forall v_k \in topk$
3. $V_c = topk$
4. **[Connection]**
5. $t_{min} = t_c$
6. $v_{min} = None$
7. **for each** $v_k \in V_c$ **do**
8.    **if** $t_i$ such that $edge(v_k)$ at $t_i = edge(v_{k_{t_c}})$ and $t_i < t_{\min}$ **then**
9.       $t_{min} = t_i$
10.      $v_{min} = v_k$
11.    **end if**
12.    **if** $v_{min} == null$ **then**
13.      $V_c =$ outgoing edges of $v_1$
14.      **goto [Connection]**
15.    **end if**
16. **end for**
17. Remove edge from $edge(v_{min_{t_c}})$ with the earliest timestamp
18. Add edge $(v_{min}, v_i)$ with timestamp $t_c$
19. $push(v_{min}) + +$
20. $prev(v) = v_{min}$
21. $v_{path} = prev(v)$
22. **while** $v_{path} \neq v_o$ **do**
23.    Add $t_c$ to $active(v_{path})$
24.    $v_{path} = prev(v_{path})$
25. **end while**
26. Add $t_c$ to $active(v_o)$

**Algorithm 2** Timestamp-limited search on the graph index

**Input**: Graph index $G$; timestamps to search $T = \{t_1, t_2, ... \}$; origin $v_o$, query vector $v_q$
**Output**: Top $K$ candidates for query $topk$

1. Initialize a binary min-heap as a priority queue $queue$ and a hash set $visited$. Construct an empty binary max-heap as a priority queue $topk$.
2. $queue \leftarrow (distance(v_o, v_q), v_o)$
3. **while** $queue \neq \emptyset$ **do**
4.    $(now\_dist, now\_vector) \leftarrow queue.pop\_min()$
5.    **if** $t_{now\_vector} \in T$ **then**
6.      **if** $topk$.size $=$ K **and** $topk$.max_dist() $\leq now\_dist$ **then**
7.        **break**
8.      **else**
9.        $topk$.push_heap($(now\_dist, now\_vector)$)
10.      **end if**
11.    **end if**
12.    **for each** $edge_{t_i} \in edge(now\_vector_{t_i})$ of $now\_vector$ where $t_i \in active(now\_vector)$ and $t_i \in T$ **do**
13.      **if** $(now\_vector, v) = edge_{t_i}$ **and** $t_i \in active(v)$ **then**
14.        **if** $visited$.exist($v$) $\neq$ **true then**
15.          $d \leftarrow distance(v_q, v)$
16.          $visited$.insert($v$)
17.          $queue$.push_heap($(d,v)$)
18.        **end if**
19.      **end if**
20.    **end for**
21. **end while**
22. **return** $topk$

Finally, we recursively walk backward along the $prev$ chain (i.e., $prev(prev(\ldots prev(v))))$, and for each vertex $v_{path}$ along it, the current timestamp $t_c$ is added to its $active(v_{path})$ set. This ensures that every node along at least one path to $v$ from the origin on $G$ is active in queries constrained to $t_c$. This mechanism guarantees that every vertex can be reached from the origin node (where all queries are initiated), at every timestamp in which it is active, without violating edge limits or requiring reconstruction of the index. The extension of this procedure to multiple timestamps is discussed in Section 3.2.

## 3.2 VERSIONING

The construction process as described in section 3.1 is designed to naturally integrate timestamp data into the graph, allowing for efficient search over flexible time ranges over the graph.

Specifically, each node maintains a versioned data structure that can quickly yield only those edges valid within a given timestamp range, and whether said node is relevant to any given timestamp. This avoids post-filtering invalid results and eliminates the need to maintain multiple time-specific indexes. The graph thereby serves as a temporally integrated index that can be traversed directly to retrieve time-consistent neighbors.

The graph building process for a multi-timestamp dataset as described in Algorithm 1 is demonstrated in Figure 2 (Figure 7 in the Appendix shows the "effective graph" for each timestamp). It should be noted that any edges that are pushed out by a future $prev(v)$ update are still valid for any timestamps after their initial creation and until their removal. For a search involving multiple timestamps (Algorithm 2), the effective graph can be considered to be a combination of the relevant timestamp graphs (see Figure 7e in the Appendix for a detailed demonstration).

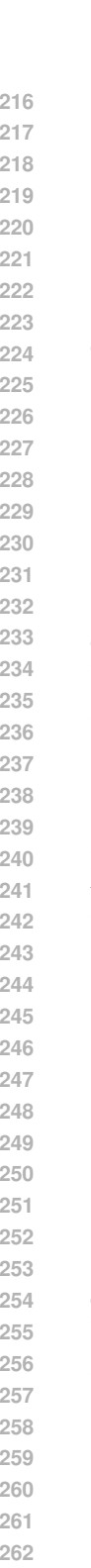
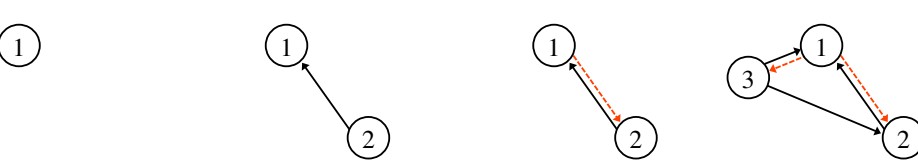
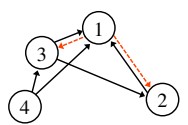
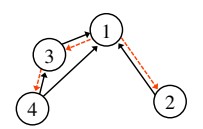
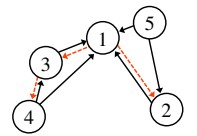
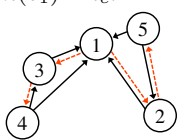

(a) Node 1 ($v_1$) is inserted and assigned as origin.

(b) Node 2 ($v_2$) is inserted. As only one other node is present, steps 1 and 2 in Algorithm 1 add edge ($v_2, v_1$) (black line).

(c) As node 1 is the closest preexisting neighbor to node 2, an edge ($v_1, v_2$) (red dotted line) is added and node 1 is assigned to $prev(v_2)$.

(d) The same process as (b) and (c) occurs for node 3. Node 1 is assigned to $prev(v_3)$ as it is closer to node 3. Note that now $push(v_1) = l_e$.

(e) The initial edge addition process for node 4. AS $l_e = 2$, edges are limited to the two nodes closer to $v_4$ ($v_1, v_2$).

(f) The $prev(v_4)$ assignment and connection process. as $v_3$ is the closest node to $v_4$, the edge ($v_3, v_4$) is added. To accommodate this, one of the existing edges of $v_3$, ($v_3, v_2$) is removed.

(g) The initial edge addition process for node 5. Note that while $v_1$ is the closest node, $push(v_1) = l_e$, and thus is not able to accommodate additional edges.

(h) The $prev(v_5)$ assignment and connection process. As $v_1$ is unavailable, the search extends to other neighbors of $v_5$ — and $v_2$ is thus selected to be $prev(v_5)$.

Figure 1: Graph construction process (Algorithm 1) for a single timestamp with 5 vectors and $l_e = 2$, which is for demonstrative purposes. In practice $l_e$ will be significantly larger. Note that all points on the graph can be reached from the origin ($v_1$) by only the $prev(v)$-enforced edges (dotted red).

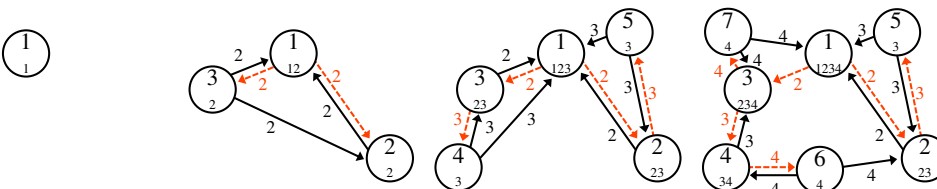

(a) Timestamp 1, with only the origin (Node 1 (or $v_1$)) added. The node is active in timestamp 1, as indicated by the number 1 below the node number.

(b) Timestamp 2 with nodes 2 and 3. Note that all edges are for timestamp 2, and node 1 is activated for timestamp 2 (as expected for the origin).

(c) Timestamp 3 with nodes 4 and 5. As $prev(v_4) = v_3$ and $prev(v_5) = v_2$, nodes 2 and 3 are also active for timestamp 3. Edge ($v_3, v_2$), which was present in timestamp 2, is no longer present.

(d) Timestamp 4 with nodes 6 and 7. Note that while node 2 has an incoming edge for this timestamp, $prev(v_6) \neq 2$ (as shown by lack of a pushed outgoing edge from node 2), and thus node 4, not 2, is active in timestamp 4.

Figure 2: Graph construction process (Algorithm 1) for 7 nodes over 4 timestamps (1 on timestamp 1, 2 and 3 on timestamp 2, 4 and 5 on timestamp 3, and 6 and 7 on timestamp 4). Nodes 1-5 are the same as in Figure 1, other than the timestamps being spread out. The smaller digits below the node number indicates which timestamps the node is active, i.e. $active(v)$. The number next to each edge indicates the timestamp said edge corresponds to. Edges that are pushed out with increasing timestamp (e.g. the edge ($v_3, v_4$) present in (c) but not (d) is still in timestamp 3 — the fact that that it was pushed out in timestamp 4 is conserved in the graph) are not featured in further timestamps.

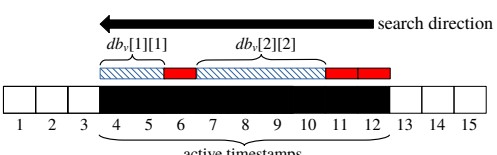

Figure 3: An example edge aggregation using the database as constructed by Algorithm 3, for timestamps [4–12] inclusive (excluding 6) and $n_s$ of 5. $n_s$ of 5 means that $db_v[1]$ is based on timestamp $1 \times n_s = 5$, $db_v[2] = 10$, and so on. The search proceeds as follows; timestamps 12, 11 are checked as individual timestamps, and a greedy search on the first valid database timestamp, 10, fetches $db_v[2][2]$ ($db_v[2][3]$ includes timestamp 6, which is not included in the search) which covers timestamps [7–10]. 6 is another individual timestamp, and $db_v[1][1]$ covers the remaining two timestamps [4–5]. Note that this search only needs to be performed once for any batch query w.r.t. the same timestamp range.

**Algorithm 3** Sparse edge database construction

**Input**: node $v$ (and corresponding sparse edge database $db_v$ and timestamp of previous update $t_{db_v}$); timestamp $t_s$ where $n_s$ divides $t_s$
**Output**: updated sparse edge database $db'_v$

1. **if** $t_s = t_{db_v}$ **then**
2.     **return**
3. **end if**
4. **for** $(t_i = t_{db_v} + n_s; t_i \leq t_s; t_i = t_i + n_s)$ **do**
5.     Determine the largest integer $j_{max}$ such that $2^{j_{max}} \leq t_s$
6.     **for** $(j = 0; j \leq j_{max} - 1; j\text{++})$ **do**
7.         **if** j = 0 **then**
8.             $db_v[t_i/t_s][0] = edge(v_{t_i})$
9.         **else**
10.             $db_v[t_i/t_s][j] = \sum edge(v_{t_k}) \quad \forall t_k$ where $t_i - 2^j < t_k \leq t_i$
11.         **end if**
12.     **end for**
13. **end for**

The search process itself proceeds similarly to a standard proximity graph search, other than a check for valid range timestamps, as shown in Algorithm 2 (demonstrated graphically in Figure 8 in the Appendix for the graph as described in Figure 2 and 7).

## 3.3 EDGE DATABASE

A very common use case for such time-based searches is searches w.r.t. a particular contiguous range of time (Zeitun et al., 2023; Zhang et al., 2022). In cases where timestamps are relatively fine-grained compared to the wanted timeframe, the required computational effort to aggregate the outgoing edges (Line 12 in Algorithm 2, or $\sum edge(v_{t_i})\mid_{t_i \in T_q}$ where $T_q$ is the desired range of timestamps) for any active node $v$ can be substantial. To address this, we describe a sparse table structure to reduce the computational cost of range aggregation during search, motivated by the structural similarity between this process and the classic range minimum query (RMQ) problem (Baumstark et al., 2017), and adopt similar construction techniques.

The process for the structure is as follows: for any timestamp $t_s$, if $t_s$ is divisible by the spacing parameter $n_s$, a positive integer set during the initial construction of each graph (i.e. $n_s \mid t_s$), Algorithm 3 is applied to each node active (whether by insertion or from a $prev(v)$ call) on said timestamp. This builds a database of periodic edge aggregations per node. As Algorithm 3 acts backward (i.e. each call to it in a timestamp $t_s$ does not involve any future timestamps) means that once a database entry for a particular node and timestamp has been established, said entry is guaranteed to be static regardless of any future updates to the graph. For any timestamp-range-based search, a greedy search on the available aggregations w.r.t. the timestamp range is performed (starting from the latest timestamp) to find appropriate aggregations and remaining timestamps (Figure 3).

While one of the main advantages of TiGER is its ability to smoothly handle noncontiguous timestamps, the edge database allows for additional speedup in contiguous timestamps (which, as stated previously, is a common use case), or cases where the query involves a discrete set of contiguous timestamps (e.g. every Friday in a dataset with timestamps on an hourly basis).

## 4 EXPERIMENTS

### 4.1 COMPARISON WITH BASELINES

To evaluate the performance of TiGER in timestamp-based dynamic workloads, we compare it against HNSW, a widely recognized state-of-the-art ANN graph-based method (Pham & Liu, 2022;

Rahman & Tesic, 2022; Ren et al., 2020). The HNSW implementation is based on its original C++ version (Malkov, 2023). We note that the broader challenges in employing other methods for meaningful comparison for this workload are discussed in Section 5. Experiments are conducted on a Linux server computer with Ubuntu 20.04.6, running on kernel version 5.4.0. Two Intel(R) Xeon(R) Gold 6438N CPUs with a clock speed of 2.00GHz, each with 32 cores and 64 threads for a total of 128 threads, are used. The system has a total memory capacity of 1.5TB. The TiGER algorithm has been implemented in C++17 with a Python interface. The code is compiled using g++-11.4.0 with "O3" optimization enabled. We incorporate two filtering strategies—post-filtering and pre-filtering—as outlined in Section 1.

In the **post-filtering** approach, filtering is integrated into the search process. Specifically, constraints are enforced during the insertion of nodes into the *topk* priority queue, ensuring that only nodes satisfying the timestamp constraints are considered (Zhao et al., 2020). The graph construction parameters follow the default settings from the original implementation, and the queries per second (QPS) vs. recall tradeoff is analyzed by varying the maximum number of vectors traversed that correspond to the timestamp range in question.

In the **pre-filtering** approach, a separate graph $G_t$ is built for each timestamp $t \in T$, where $T$ represents the set of timestamps in the dataset. During a search, we query only the graphs $G_t$ corresponding to timestamps $t \in T_q$, where $T_q \subseteq T$ denotes the timestamps satisfying the given constraints. The results from these individual searches are combined using parallel k-way merging (Lee & Batcher, 1995), with early stopping techniques as with the Best Position Algorithm (Akbarinia et al., 2007) to optimize performance. The QPS vs. recall curve is generated by varying the size of the *topk* lists obtained from each graph. Values of individual *topk* below final *topk* are also tested (as the true topk is likely to be distributed among $T_q$ and thus the full textit*topk* is likely to be not necessary for individual searches).

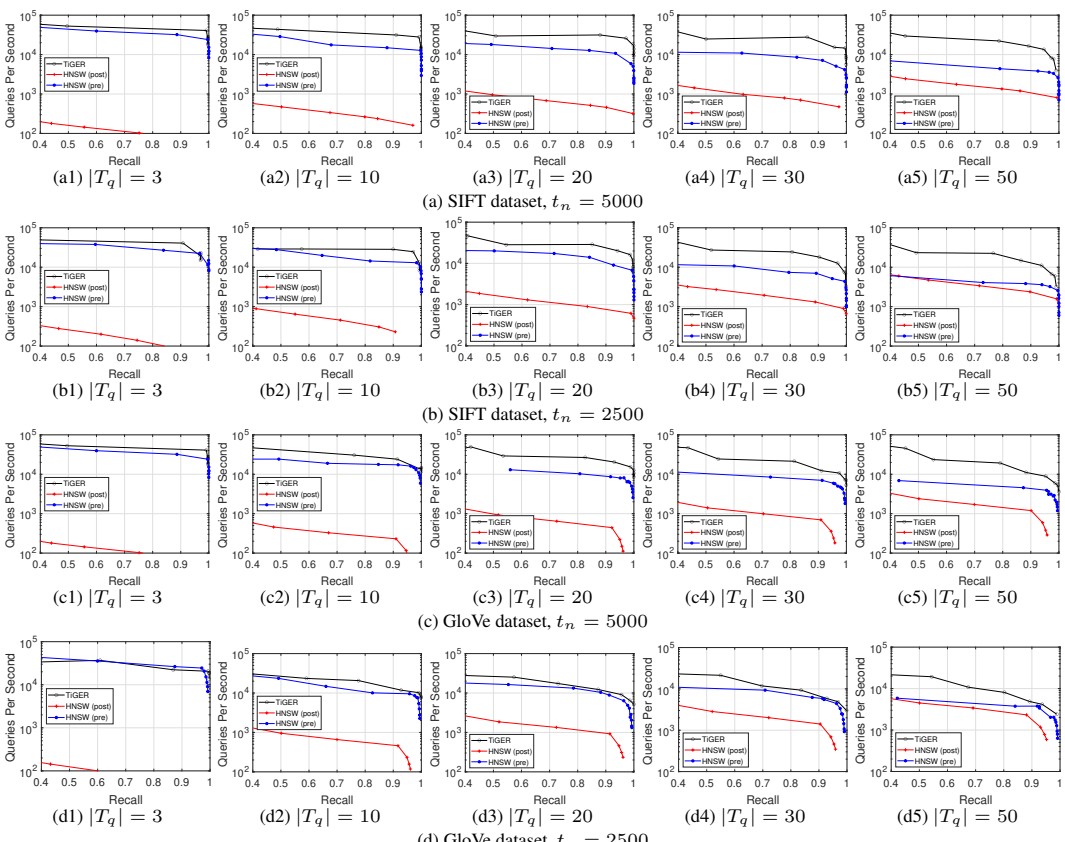

Figure 4: Recall vs. queries per second (QPS) for contiguous $T_q$ for TiGER and baselines for $k = 100$. "HNSW (pre)" indicates pre-filtering HNSW (i.e. final *topk* is produced by merging of per-timestamp HNSW search results) and "HNSW (post)" indicates post-filtering HNSW (i.e. *topk* is produced by filtering out vectors that do not correspond to $T_q$ during search). TiGER maintains a lead over baselines over a wide range of $|T_q|$.

## 4.2 Workload Simulation

TiGER is designed to operate on time-based datasets with dynamic insertions, accommodating continuous updates without requiring graph reconstruction. To simulate such a workload, we apply the following process to standard ANN datasets:

1. Divide the dataset $D$ into $n$ artificial timestamps, as $T = \{t_1, t_2, \ldots, t_n\}$, in assumed chronological order.
2. Construct an initial proximity graph $G$ using the vectors associated with the earliest timestamp, $t_1$.
3. Sequentially insert vectors associated with subsequent timestamps in ascending order.
4. Once all vectors up to $t_n$ are inserted, perform a search on the graph to retrieve the *topk* nearest neighbors for a set of query vectors $Q$, constrained by timestamps $T_q \subseteq T$.

We apply the workload to vector datasets that are standard in ANN literature; namely, we use the SIFT 1M dataset (Jégou et al., 2011), a 128-dimensional dataset consisting of 1 million vectors, and the GloVe-100 (Pennington et al., 2014) dataset with 100 dimensions. We employ the query vector set provided with each dataset, and set $k = 100$.

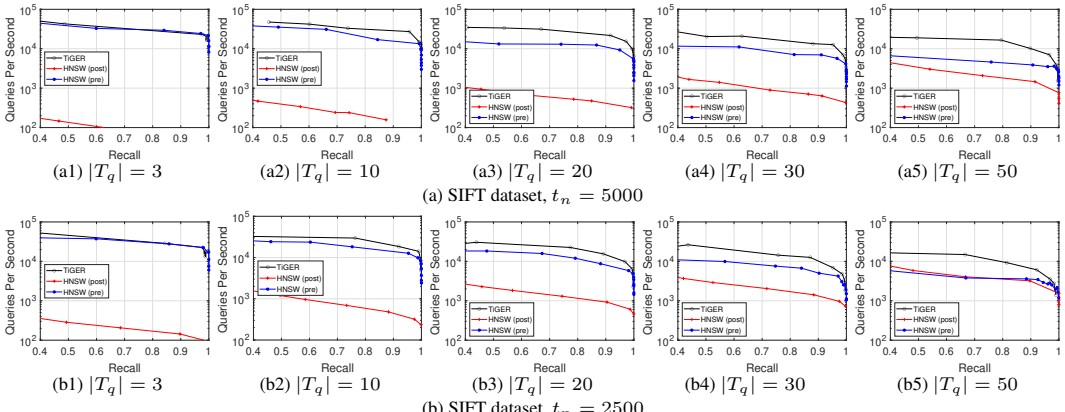

Figure 5: Recall vs. queries per second (QPS) for discrete $T_q$ sets (i.e. no sequential timestamps in set) for TiGER and baselines for $k = 100$. As with Figure 4, "HNSW (pre)" indicates pre-filtering HNSW and "HNSW (post)" post-filtering HNSW. TiGER maintains a lead (although slightly less pronounced than for Figure 4) over baselines over a wide range of $|T_q|$.

For $t_n$ and $T_q$, our settings are as follows: $t_n = 2500, 5000$ and $T_q = 3, 10, 20, 30, 50$. We vary $t_n$ to account for different per-timestamp dataset sizes and their effect on the results. We also vary $T_q$ sizes in order to cover both small sets (i.e. tighter filter) where pre-filtering would be effective (due to the lower number of arrays required to merge) and large sets (i.e. looser filter) where post-filtering would be effective (as proportionally more checked points are valid, fewer traversals are expected to encounter a valid vector and in turn form a complete *topk* queue).

The experimental results for contiguous $T_q$ (e.g. $T_q = \{13, 14, 15\}$ for $|T_q| = 3$) are shown in Figure 4. While pre- and post-filtering act as expected, with larger $T_q$ (i.e., wider filters) improving post-filtering, and pre-filtering dropping off at high recall, TiGER maintains a lead over either method in QPS vs. recall consistently over the different filter lengths.

Discrete $T_q$ are also applied to discern the effect of non-contiguous $T_q$ (for which the edge database described in section 3.3 would be ineffective). For such $T_q$, we apply a filter in which each $t \in T_q$ is at least spaced by 1 from all other $t \in T_q$ (e.g. $T_q = \{26, 28, 30\}$ for $|T_q| = 3$). This prohibits the edge database from fetching any compacted ranges from its search. The results for SIFT dataset are shown in Figure 5, which demonstrates that the gains of TiGER as shown in Figure 4 are still present.

## 5 Related Work

Efficient approximate nearest neighbor search with additional constraints such as numeric ranges has received significant attention in recent years. Various methods have been proposed to address

the challenges associated with integrating these constraints into proximity graphs. We discuss the relevant studies in this section.

Segment Graph for Range-Filtering ANNS (SeRF) (Zuo et al., 2024) introduces segment graphs in which multiple indices for contiguous numeric ranges are compressed into a single structure. By annotating edges with range validity, SeRF enables efficient traversal for contiguous range queries. However, it does not natively support disjoint ranges, requiring multiple searches or preprocessing steps for such constraints. Furthermore, SeRF lacks full support for dynamic updates, necessitating substantial reconstruction when new data or ranges are added to the dataset.

Unified Navigating Graph (UNG) (Cai et al., 2024) employs a Label Navigating Graph (LNG) to organize data hierarchically based on label containment relationships. This enables efficient filtered ANNS for categorical or hierarchical labels. However, UNG also struggles with dynamic updates: adding new data often requires cross-range reconstruction for integrity of hierarchical relationships.

iRangeGraph (Xu et al., 2024) addresses range filtering by precomputing elemental graphs for specific ranges and dynamically merging them during query execution. This approach achieves a balance between memory efficiency and query performance for continuous ranges. However, disjoint ranges require combining multiple elemental graphs with substantial query-time overhead. iRangeGraph also lacks inherent support for dynamic updates, making it less suitable for evolving datasets.

Filtered-DiskANN (Gollapudi et al., 2023) extends the Vamana proximity graph to support label-based filtering. It introduces FilteredVamana, which incrementally builds a graph by pruning connections based on filter-specific constraints, and StitchedVamana, which creates separate graphs for each filter and merges them into a unified structure. While these methods enable efficient queries for predefined filters, they are difficult to maintain for frequently evolving filters. StitchedVamana, in particular, necessitates costly graph rebuilding or re-stitching to handle dynamic updates.

Native Hybrid Query (NHQ) (Wang et al., 2022b) aims to address queries by combining vector similarity with attribute-based filtering. NHQ processes such queries using a composite proximity graph and a fusion distance metric, which integrates feature similarity and attribute compatibility. This metric guides a joint pruning strategy that eliminates candidates failing either constraint during graph traversal. To handle range-based constraints, NHQ either has to: perform separate pruning and merging for each range, which can increase query latency, or predefine connectivity for all possible ranges during construction. NHQ also relies on a predefined fusion distance threshold to determine graph connectivity. Adding new ranges not present during initial construction requires modifying the composite index along with the changing fusion distance threshold, further complicating updates. This limitation reduces its adaptability to datasets with evolving constraints or highly dynamic ranges.

DIGRA (Jiang et al., 2025) combines multi-way tree structures with navigable small-world (NSW) graphs to support efficient range-aware queries. Unlike many earlier approaches, DIGRA provides native support for dynamic updating. However, its update operations are currently restricted to single-threaded execution, limiting scalability in high-throughput environments.

## 6 CONCLUSIONS

The rise of applications requiring time-sensitive ANN searches has highlighted significant limitations in existing graph-based methods. However, current approaches have been computationally inefficient or problematic w.r.t. dynamic updates and/or noncontiguous filters.

To this end, we introduce TiGER (Time-Integrated Graph for Efficient Retrieval), a novel graph-based framework specifically designed to efficiently manage range-filtered approximate nearest neighbor (RFANN) searches with time-based constraints in large, dynamic datasets. TiGER leverages a unified proximity graph supplemented with versioned connectivity metadata, eliminating the need for post- or pre-filtering strategies. This ensures both scalability and adaptability while enabling seamless dynamic updates.

Empirical evaluations across standard ANN benchmarks, demonstrate the effectiveness of TiGER. Our results show up to a 5x improvement in query performance in a wide range of filters compared to baselines such as HNSW with both pre-filtering and post-filtering strategies. This consistent advantage highlights TiGER's ability to balance recall and query speed across diverse workloads while maintaining adaptability to evolving datasets.

## ETHICS STATEMENT

This work does not involve human subjects, personally identifiable information, or the scraping of private data. Experiments use only public vector datasets. We do not anticipate any ethical issues that arise from this work beyond those associated with the standard use of databases.

## REPRODUCIBILITY STATEMENT

We present our approach with both conceptual and algorithmic detail in Section 3, and clearly document the experimental procedures in Section 4. All datasets used in our experiments are public and well-documented. We also provide an anonymous repository for this work in Section D of the Appendix.

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

# APPENDIX

# A  ADDITIONAL EXPERIMENTS

## A.1  EDGE DATABASE

To quantify the effects of the edge database on search speed as described in Section 3.3 and indirectly compared in Figures 4 and 5, we evaluate TiGER's search speeds both with and without utilizing the edge database. For each filter with varying $T_q$ sizes on the same graph, we conduct two searches: one employing the standard edge database search and another where edge validity for each range is determined by brute-forcing through each $t \in T_q$. We also apply this process for both contiguous and discrete $T_q$, the latter of which the current edge database is not able to fetch useful aggregations and is thus expected to behave in the same way as TiGER without the edge database applied.

The results are presented in Figure 6. Overall the performance is as expected, with a general visible gain seen throughout the range of $T_q$ for contiguous timestamp filters. This gain also substantially increases with increasing $T_q$, as the edge database can compact an increasing number of timestamps. Searches with discrete $T_q$ shows little visible difference with or without the edge database.

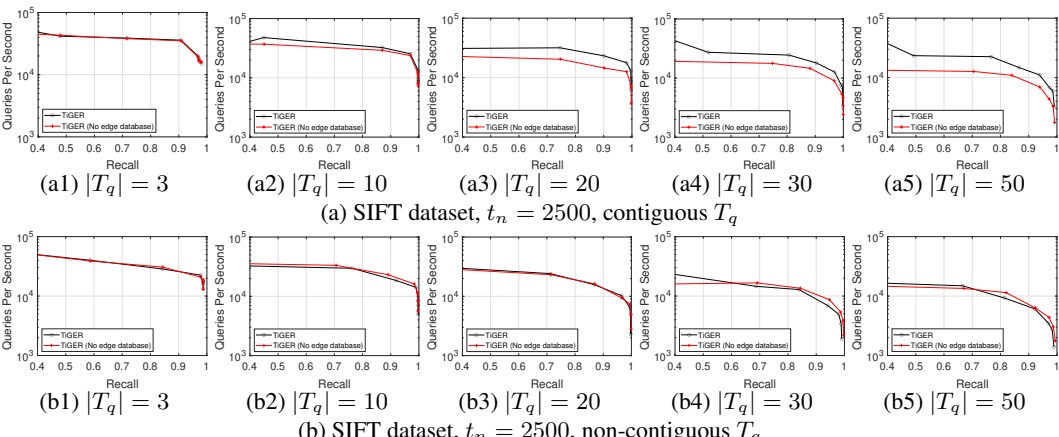

  (a1) $|T_q| = 3$      (a2) $|T_q| = 10$      (a3) $|T_q| = 20$      (a4) $|T_q| = 30$      (a5) $|T_q| = 50$

(a) SIFT dataset, $t_n = 2500$, contiguous $T_q$

  (b1) $|T_q| = 3$      (b2) $|T_q| = 10$      (b3) $|T_q| = 20$      (b4) $|T_q| = 30$      (b5) $|T_q| = 50$

(b) SIFT dataset, $t_n = 2500$, non-contiguous $T_q$

Figure 6: Comparison of TiGER search speeds with and without the edge database as described in section 3 for contiguous and discrete $T_q$ for $k = 100$. contiguous $T_q$ filters show visible improvement with the application of the edge database at higher $T_q$, with the gap increasing with larger $T_q$. With discrete filters, no substantial gap is present at any size of $T_q$.

# B  ADDITIONAL FIGURES

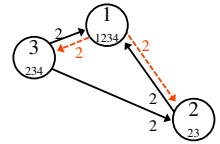

(a) Effective graph of timestamp 1.

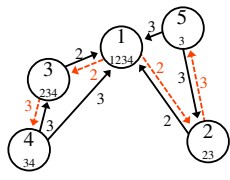

(b) Effective graph of timestamp 2. The edge $(v_3, v_2)$ will be pushed out in timestamp 3.

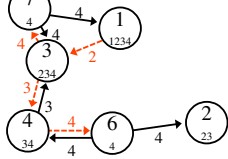

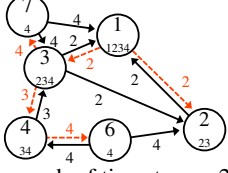

(c) Effective graph of timestamp 3. the edge $(v_4, v_1)$ will be pushed out in timestamp 4.

(d) Effective graph of timestamp 4. Node 5 is not active. node 2 is also not active, although it is shown due to its connection with a timestamp 4 edge $(v_6, v_2)$.)

(e) Effective graph of timestamps 2 and 4 combined. As node 5 is only active on timestamp 3, it is not present. Additionally, the edge $(v_3, v_2)$, which has been pushed out in timestamps 3 and 4, is present due to its presence in timestamp 2.

Figure 7: The effective graphs for each timestamp w.r.t. construction process as in Figure 2 ((a)-(d))) and (e) the effective graph for a search on timestamps 2 and 4. As node 5 is only present and/or active on timestamp 3, it does not appear on the effective combined timestamp graph.

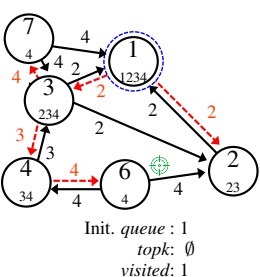

Init. *queue* : 1
*topk*: ∅
*visited*: 1

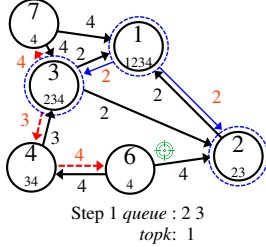

Step 1 *queue* : 2 3
*topk*: 1
*visited*: 1 2 3

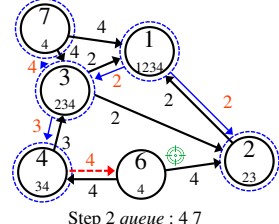

Step 2 *queue* : 4 7
*topk*: 2 1
*visited*: 1 2 3 4 7

(a) Start of search (w.r.t. vector marked with green crosshair) on the graph, from origin $(v_1)$. Checked nodes are marked with blue dotted circles.

(b) First step of search. 1 is been assigned to *topk*. Node 2 and 3 are active in timestamp 2 and are placed on the queue.

(c) Second step of search. Node 2 yields no new connections. Node 3 is then popped and its edges evaluated, which adds 4 and 7 (from timestamp 4) to the queue.

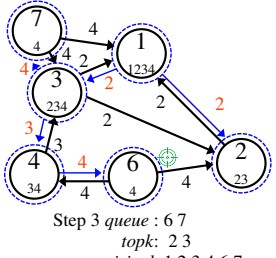

Step 3 *queue* : 6 7
*topk*: 2 3
*visited*: 1 2 3 4 6 7

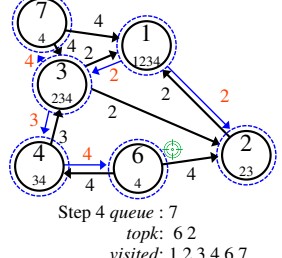

Step 4 *queue* : 7
*topk*: 6 2
*visited*: 1 2 3 4 6 7

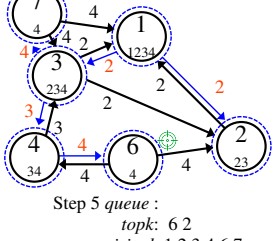

Step 5 *queue* :
*topk*: 6 2
*visited*: 1 2 3 4 6 7

(d) Third step of search. Node 4 is popped but not evaluated due to $t_{v_4} = 3 \notin \{2, 4\}$ (but its edges are traversed).

(e) Fourth step of search. Node 6 is evaluated.

(f) End of search. the queue is empty, and thus the search is ended with a *topk* of 6, 2.

Figure 8: A search on the graph index for $T_q = \{2, 4\}$ as constructed in Figure 2 for a *topk* limit of 2. The target query vector is marked with a green crosshair. paths and nodes traversed are marked in blue. Note that node $v_4$ is not evaluated as $t_{v_4} = 3 \notin \{2, 4\}$, but as $\exists x \in active(v_4) = \{3, 4\}$ for which $x \in \{2, 4\}$ (as $active(v_4) = \{3, 4\}$), is assessed for valid edges, bridging nodes 3 and 6.

## C    USE OF LARGE LANGUAGE MODELS

Large Language Models were used to aid in debugging, as well as to polish the grammar and clarity of the text in this paper. The responsibility for all content within this paper lies solely with the authors.

## D    SUPPLEMENTARY MATERIALS

We provide an anonymous repository corresponding to this work for reproducibility: `https://anonymous.4open.science/r/TiGER-CF36`

