# OpenReview forum: "A Versioned Unified Graph Index for Dynamic Timestamp-Aware Nearest Neighbor Search"
_ICLR.cc/2026/Conference — ICLR 2026 Conference Withdrawn Submission_

### Official Review · Reviewer_GWcD · 2025-11-01

**Soundness:** 2
**Presentation:** 2
**Contribution:** 2
**Rating:** 2
**Confidence:** 4

**Summary:**

This paper proposes a new method for ANN search over temporal graphs. The paper presents the proposed algorithm and some experiments without a clear problem statement, complexity analysis, or dataset description.

**Strengths:**

The research problem is practically interesting.

**Weaknesses:**

1. The writing could be improved. I would like to see a more formal problem statement for such a type of work, since temporal queries do have a large number of variations.
2. Incomplete literature review. There are a lot of studies on querying versioned graphs (maybe under different names such as dynamic graph, temporal graph, etc.). Many of them date back to the 2000s. Although they may not be suitable for embeddings, it is still necessary to mention them, explaining their deficiencies and emphasizing the novelties of this research.
3. Algorithms 1 & 2 are over-lengthy with unnecessary programming-level details.
4. The proposed method is for discrete timestamps. In this work, by “continuous”, the authors actually mean consecutive timestamps. However, in practice, time is really continuous. Discretizing time may lead to a huge number of timestamps, making the algorithm computationally prohibitive.
5. There’s no theoretical (in particular, complexity) analysis, which I believe is unacceptable for such a type of work.
6. No dataset description in Sec. 4 (content, size, etc.). I cannot understand the experiments.

**Questions:**

See detailed comments.

---

### Official Review · Reviewer_C2Gy · 2025-11-01

**Soundness:** 3
**Presentation:** 2
**Contribution:** 3
**Rating:** 4
**Confidence:** 4

**Summary:**

The paper provides a procedure to solve the ANNS problem with timestamp constraints. This problem is indeed important. It provides some novel ideas on how to handle timestamp filters, timestamp range filters and also dynamic insertions.

However, I feel, the comparison to previous baselines should be done in experiments section in a much thorough manner. The value of current contributions is not evident from the experiments performed. As this is an empirical paper, I feel changing some of the algorithms from the prior work for the problem studied is important to have meaningful baselines to do comparison against.

I liked the ideas presented in the paper on handling timestamp filters and also range filters. However, after reading the paper, I don't yet know how good the algorithm is. More experiments should be done to convince me the value of the algorithm presented. The authors should also talk about space usage, build times and other important metrics. The shortcomings of the proposed approach should also to be discussed in the paper.

Overall, the paper has promising ideas. Further revision will make it stronger and worthy to be accepted at venues like ICLR. I am leaning in the middle for this paper: that is, neither accept or reject.

**Strengths:**

1) To handle timestep filters, I liked that the authors introduced timesteps for edges and they maintain graph connectivity among vertices associated with a particular time step.

2) The idea for handling range filters is also nice.

**Weaknesses:**

1) I feel the writeup could be done better. For instance, in your write up, please separate problem statement from your procedure to solve it. You could clearly state what the problem statement is.

2) The experiment section also looks very primitive. Not lot of effort in adding other baselines has been done. For non-range filter queries, you should compare your approach with existing methods and show that yours does well.

**Questions:**

1) Would you please state the problem statement formally somewhere? I mean, mathematically, what is the problem that you are trying to solve. Also by dynamic, you mean only insertions? Are there no deletions?

Are documents arriving in an online fasshion? How is a timestep associated with a document defined? Is it given as input to the index algorithm? In particular, please explain the problem statement exactly. Is t_{v} and active(v) part of inputs to the index building algorithm?


2) Why is Filtered-DiskANN not part of your baselines? It can handle dynamic inserts I think. This is just one of the algorithms which can be directly applied in your setting.

3) If there are baselines which cannot handle insertions but can handle filters, can we just compare your approach to these baselines where there are no new insertions?

---

### Official Review · Reviewer_8zfo · 2025-11-02

**Soundness:** 2
**Presentation:** 1
**Contribution:** 1
**Rating:** 2
**Confidence:** 5

**Summary:**

This paper proposes a time-constrained approximate nearest neighbor search method for data with assigned timestamps. The proposed approach extends graph-based neighbor search methods by associating each node with timestamp information and employing various auxiliary structures to enable timestamp-aware search.

**Strengths:**

The focus of this paper, "Approximate Nearest Neighbor Search Considering Timestamps," is important and convincing. Timestamped data are ubiquitous, yet most existing approximate nearest neighbor search methods do not provide explicit solutions for handling them. There is a strong demand for data structures that consider timestamps. The formulation presented in Sec. 1, "Our Approach," for expressing this problem is well-reasoned and logically consistent.

**Weaknesses:**

This paper addresses an interesting research topic. However, there are several points where the descriptions are insufficient, as detailed below. In its current form, I do not believe the manuscript is ready for acceptance.

## Complexity of Explanation

Above all, the proposed method is complex and difficult to understand. The authors make a sincere effort to explain the method clearly through pseudocode (Algorithms 1, 2, and 3) and simple figures (Figs. 1, 2, and 3), but even so, the method remains difficult to grasp. Due to this complexity, it becomes unclear whether the proposed approach is indeed effective.

## Deficiencies in Notation

Related to the above, there is significant room for improvement in the paper's notation.

- First, there are too many subscripted variables, which significantly reduces readability. For example, `v_k_t_c` in Algorithm 1, line 8, has three levels of subscripts and is very hard to interpret.
- It is recommended to use Roman type for ordinary English words. For instance, in Algorithm 1, line 17, the notation `v_min_t_c` would be more readable if written as `v_\mathrm{min}`.
- In the pseudocode, the style of description should be unified, whether to use mathematical notation or pseudocode-style notation. For example, the `=` symbol is used for assignment in Algorithm 1 (e.g., line 5), but for equality in Algorithm 2 (e.g., line 6). This inconsistency can be considered an error.
- In Figure 2, both timestamps and node indices are represented by small integers, which is confusing. It would be clearer to use a different font or distinct numerical ranges for these two types of values.

## Lack of Baselines

The paper lacks sufficient baselines for comparison. The only comparison is with HNSW enhanced by simple pre- and post-processing, but it is obvious that such a setup would perform poorly under extreme parameter configurations. It would be preferable to include comparisons with other related methods mentioned in Section 5.

## No Description of Memory Consumption

The paper does not provide a concrete discussion of memory consumption. Data structures for neighbor search must be evaluated in terms of the trade-offs among accuracy, runtime, and memory. Without memory usage analysis, the discussion remains incomplete (for instance, if unlimited memory were available, entirely different and more effective methods might be feasible). Specifically, the structure `active(v)` likely maintains a set (e.g., `std::set`) for each node, which requires memory space. But such considerations are not discussed in the paper.

**Questions:**

In Algorithm 1, the output is listed as only the updated index $G'$; however, at a minimum, a data structure such as `active(v)` should also be required. How many such hidden data structures are there?

---

### Official Review · Reviewer_AWwp · 2025-11-04

**Soundness:** 2
**Presentation:** 3
**Contribution:** 1
**Rating:** 2
**Confidence:** 4

**Summary:**

The authors present a TiGER model to support temporal filtering during search while enabling dynamic updates. Experimental study shows that TiGER achieves 5x improvement over HNSW (pre) and HNSW (post).

**Strengths:**

Strength

S1. This paper is self-contained and easy to follow.

S2. The problem may be interesting for some readers.

**Weaknesses:**

Weakness

W1. Motivation is not clear.

W2. The experimental setup is not convincing.

W3. Novality is not clear.

Detailed comments

D1. This paper proposes a vector retrieval algorithm designed for temporal data to handle dynamic updates. First, are there real-world applications that support this specific scenario? Secondly, in practical applications today, vector datasets often reach billions of entities with substantial daily updates and high dynamism. It remains unclear whether the proposed method can support such a scenario with intense fluctuations.

D2. The paper lacks a theoretical analysis of the method's effectiveness. Merely describing the algorithm's logic and demonstrating its efficacy through experiments is insufficient.

D3. The authors need to explain why HNSW was chosen from the many available ANNS algorithms. Furthermore, comparing only with HNSW is far from sufficient; a comparison with more algorithms is necessary.

D4. To strengthen the validation, the experimental evaluation should be expanded. It would be beneficial to include datasets exhibiting different update patterns (e.g., high-frequency vs. low-frequency) and at least one large-scale (hundreds of millions) dataset. This would more convincingly demonstrate the robustness and scalability of the proposed method.

**Questions:**

Please refer to the weakness part.

---

### Note · Authors · 2025-11-25

**Comment:**

We would like to thank the reviewers for their time and their detailed evaluation of our submission. We appreciate the reviewers' comments on the need to strengthen the work's positioning and evaluation.

We acknowledge the feedback regarding the problem formulation and the need for a more formal theoretical analysis. We recognize that, while the heuristic and data-dependent nature of graph-based ANN methods can make deriving tight bounds challenging, providing a clearer formalization of the problem statement and discussing the memory consumption complexities are important steps we intend to consider moving forward.

Regarding the empirical study, our focus on dynamic insertions initially led us to limit comparisons to methods that natively support that workflow. However, we also appreciate the feedback regarding the exclusion of filtering baselines, such as Filtered-DiskANN, and the limitation of dataset scale, which makes it difficult to fully gauge the method's utility.

We are grateful for the constructive criticism and will take these points into careful consideration in further revision.

**Withdrawal Confirmation:**

I have read and agree with the venue's withdrawal policy on behalf of myself and my co-authors.